# Preparation of Nitrogen-Doped ZnFe₂O₄-Modified Carbon Composite and Its Collaborative Energy Storage Mechanism

Li Wang [1], Baobao Li [2], Hongyu Bai [3], Hong Ding [4], Na Xu [4], Chaofan Yin [1], Jingjing Xiong [2], Zhiwei Yang [2], Xianfa Rao [2,*] and Binbin Dong [1,*]

[1] School of Materials Science and Engineering, Henan Province International Joint Laboratory of Materials for Solar Energy Conversion and Lithium Sodium Based Battery, Luoyang Institute of Science and Technology, Luoyang 471023, China; wangli6433@sina.com (L.W.)

[2] Faculty of Materials Metallurgy and Chemistry, Engineering Research Institute, Jiangxi University of Science and Technology, Ganzhou 341000, China

[3] Yanshi Zhongyue Refractory Co., Ltd., Luoyang 471900, China

[4] Anhui Product Quality Supervision & Inspection Research Institute, Hefei 230051, China; 13956970648@139.com (H.D.)

[*] Correspondence: raoxianfa@126.com (X.R.); dongbb@lit.edu.cn (B.D.)

**Abstract:** The pyrolytic carbon of polymer adsorbent resin (SAP) is used as a waste carbon source, which can be used as a porous carbon network via pyrolysis to remove surface sodium carbonate and other substances. In this paper, a ZnFe₂O₄/nitrogen-doped porous carbon composite was prepared using the template method. Through the high-temperature carbonization of a polymer and crystallization of inorganic elements, the morphology of the composite showed uniform load characteristics. This well-defined structure and morphology facilitate the transport of Li⁺, enhance the effective contact area with the electrolyte, and provide a wealth of active sites. For the SAP-Fe/Zn anode, at a high current density of $0.1 \text{ A g}^{-1}$, the reversible capacity of the anode reached $753 \text{ mAh g}^{-1}$ after 200 cycles, showing excellent magnification performance. The final modified SAP-Fe/Zn&NC electrode had a reversible capacity of $205.6 \text{ mAh g}^{-1}$ after 1000 cycles at the high current density of $2 \text{ A g}^{-1}$, and the cycle retention rate was as high as 80.7%. The enhanced electrochemical performance can be attributed to the abundant active sites and shortened diffusion pathway of the composite. This ensures adequate conversion reactions during the Li-litization process between Zn, Fe, and Li⁺, alleviates volume expansion, and prevents comminution/aggregation during long cycles at high current densities.

**Keywords:** SAP; ZnFe₂O₄; nitrogen doping; carbon composites; lithium-ion battery

## 1. Introduction

Due to the rapid consumption of fossil fuels and the subsequent serious environmental problems, the utilization of sustainable and environmentally friendly new energy sources such as wind energy, solar energy, geothermal energy, hydropower, and tidal energy has attracted great attention in the past few decades [1–6]. However, problems such as unpredictability, capacity instability, and intermittence hinder the development and utilization of these energy sources [7–9].

The lithium-ion battery (LIB) has been regarded as one of the most outstanding technologies since it was first commercialized in the early 1990s. It has greatly reshaped our lives, and the continuous improvement in its derivatives in materials and chemistry may determine our energy future. Among the anode materials of lithium-ion batteries, the conversion anode material has a high specific capacity and a wide source of materials [10–13]. However, in the process of charging and discharging, there will be obvious volume change, which will cause the collapse of the material structure. In addition, the low charging and discharging efficiency greatly limits its application in the field of energy storage [14]. As the

cathode material of the lithium-ion battery, carbon material can provide a certain capacity. Studies have shown that nitrogen-doped carbon materials not only have high conductivity, but also enhance the mechanical properties of composites. Its flaky structure can effectively alleviate the volume effect of active materials, thus improving their electrochemical properties [15].

Polymer water-absorbent resin (SAP) can be used as a high-quality carbon skeleton of carbon-based composites by removing substances such as sodium carbonate from its surface after carbonization. The researchers modified it with transition metal oxides (iron oxide, cobalt oxide, zinc oxide) to obtain carbon-based transition metal oxide composites with excellent properties. However, when $Fe_2O_3$ is used as the anode material of a lithium-ion battery alone, the capacity of the battery is very low. Under the current density test of 100 mA $g^{-1}$, the first-cycle discharge-specific capacity is 187.5 mAh $g^{-1}$, and the first-cycle charge–discharge efficiency is 47.8%. As the anode material of the lithium-ion battery, ZnO has a high theoretical capacity of 978 mAh $g^{-1}$, but it shows poor electrochemical performance due to its low electronic conductivity and large volume change during $Li^+$ intercalation and desorption [16]. However, the existence of multiple metals can usually induce higher conductivity, because the activation energy of electron transfer between cations is lower than that of a single metal oxide structure [17]. Nanostructured $ZnFe_2O_4$ composites with good crystallinity and specific particle shape are highly needed for application in the LIB, because small particles can greatly shorten the reaction path of $Li^+$, increase the electron/ion conductance, and adapt to the volume expansion caused by charge/discharge [18]. Zn is considered to be a promising candidate to combine with $Fe_2O_3$ to form zinc ferrite ($ZnFe_2O_4$) because of its advantages of non-toxicity, environmental friendliness, stable structure, and low working potential.

Li [19] used a simple method to prepare high-performance supercapacitor electrode $ZnFe_2O_4$/NRG composites by anchoring ultra-small $ZnFe_2O_4$ nanoparticles onto nitrogen-doped reduced graphene (NRG). NRG as a substrate cannot only control the formation of nano-$ZnFe_2O_4$, but also ensure high dispersion without the agglomeration phenomenon. Thanks to this novel combination and structure, the hybrid material has a large surface area and can provide highly exposed active sites for easy electrolyte acquisition and fast electron transport. Wu [20] prepared $ZnFe_2O_4$/carbon (C) @N-doped porous carbon nanotube (NCNT) composites via one-step pyrolysis using Zn-Fe-ZIF as a precursor and a sacrificial template. The specific surface area of the $ZnFe_2O_4$/carbon (C) @N-doped porous carbon nanotube (NCNT) composites was up to 256 $m^2$ $g^{-1}$. The carbon content of the porous $ZnFe_2O_4$/C@NCNT nanocomposites was about 50%. At 100 mA $g^{-1}$ current density, the initial discharge capacity of the porous $ZnFe_2O_4$/C@NCNT nanocomposites was 2192 mAh $g^{-1}$, and the highly reversible capacity of 844 mAh $g^{-1}$ could be reached after 100 cycles.

Ding [21] and others synthesized nanostructured ternary transition metal oxide $ZnFe_2O_4$ via polymer pyrolysis, which showed high specific capacity and good cycle performance.

In this study, $ZnFe_2O_4$-coated carbon composites were synthesized through the use of a two-step pyrolysis strategy, and nitrogen-doped $ZnFe_2O_4$-modified carbon composites with different morphological characteristics were prepared using $ZnFe_2O_4$ as a template. The application of layered porosity of this multistage material can provide more active sites for lithium-ions, and the designed composite materials have large absorption and high bearing capacity. The shell is thin and there is free space inside, which further improves the circulation stability of the material.

## 2. Experimental Part

### 2.1. Preparation of Sap-Fe/Zn, SAP-Fe/Zn&N, and SAP-Fe/Zn&NC Samples

Firstly, $ZnCl_2$ and $FeCl_2 \cdot 6H_2O$ were dissolved in 200 mL deionized water at the same mass ratio; then, excessive NaOH aqueous solution was added and stirred via magnetic force at 50 °C, SAP powder was added, and it was stopped when the solution completely

became a hydrogel. This was recorded as sample 1. Then, $ZnCl_2$ and $FeCl_2 \cdot 6H_2O$ were mixed with 5 g urea in 250 mL deionized water to form a bright yellow layered turbid liquid, which was left standing for 1 h, and then this was recorded as sample 2 after a hydrogel was formed. Two groups of samples were placed in the refrigerator at $-20\,^\circ\text{C}$ and pre-frozen for 5 h. After being completely frozen, the precursor was obtained by freeze-drying for 48 h. The precursor was pulverized into powder. Then, the powder was heated to $200\,^\circ\text{C}$ at $3\,^\circ\text{C min}^{-1}$ in $O_2$ atmosphere in a tube furnace, then heated to $600\,^\circ\text{C}$ at $1\,^\circ\text{C min}^{-1}$ in nitrogen atmosphere, and cooled to room temperature. The obtained black solid was crushed, ground, and sieved, and then immersed in 100 mL (1.0 mol/L) concentrated hydrochloric acid for 30 min to corrode any unstable metal. Then, the samples were washed with ultrapure water and anhydrous ethanol in turn and dried at $60\,^\circ\text{C}$, and the obtained materials were recorded as SAP-Fe/Zn and SAP-Fe/Zn&N, respectively. Then, parts of the SAP-Fe/Zn&N samples were put into a clean porcelain boat, the SAP-Fe/Zn&N samples were placed in a tube furnace to rapidly raise the temperature to $600\,^\circ\text{C}$ in an oxygen atmosphere, and the temperature was kept for 1 min and then rapidly annealed to $500\,^\circ\text{C}$. The calcined sample was recorded as SAP-Fe/Zn&NC.

### 2.2. The Preparation of Electrochemical Performance Test

2.2.1. Electrode Anode Paste Preparation Process

1. Beating: Use electronic scales according to the active substances/acetylene black (SP)/oily adhesive (PVDF) = 8:1:1. Put an amount according to the proportion of the total weight for 1 g of powder material into the beaker anode paste mixing configuration. Firstly, dissolve 0.1 g PVDF in NMP solution in a ratio of 1:29. Secondly, place the prepared solution in a drying oven at $120\,^\circ\text{C}$ until the PVDF is completely dissolved. Then, add 0.1 g acetylene black and 0.8 g active substance. In the fourth step, add six ball mills including one of a large particle size, two of medium particle size, and three of small particle size. Wrap the small beaker with aluminum foil and seal the beaker with plastic wrap. Finally, put the beaker into the agate can and ball mill for 6~8 h at a constant speed of $18\text{ r s}^{-1}$.

2. Coating: Start the dehumidifier, wipe the knife table and the surface of the coating machine with alcohol, and adjust the knife table to zero. Keep the copper foil rough face up and open the vacuum pump, so that the copper foil closely adsorbs on the surface of the coater. Adjust the scale of the knife table to 10–12 μm, apply the pulp to the right side of the copper foil using the medication spoon, place the knife edge parallel to the right side of the pulp, start the coater, move the knife table to the left at a speed of $7\text{ mm s}^{-1}$, wipe the knife edge, and set it to zero after coating. The coated electrode sheet should be placed in the oven at $120\,^\circ\text{C}$ for more than 30 min to dry it, and the NMP should be volatilized.

3. To roll: Start to roll the machine, using alcohol to clean the roller. Press the roller to exert pressure; after drying the pole piece of the roller, measure the roller pressure; after pressure relief, wipe the roller press and roller wheel again.

4. Plate: the plate will not be kept to roll the sheet after coating, and the oriented anode plate machine using a dedicated fine structure will suppress the cathode piece into a 1.4 cm diameter size small round plate.

5. Weighing: Blunt the circular pole piece weighing, and record the data; choose a quality close to the plate and make sure that the surface level is smooth; minus the bare copper foil quality of 15.1 mg, and multiply it by 0.8; this should conclude the quality of the active material.

6. Drying: After weighing the sheet number, place it in a vacuum drying oven and adjust the temperature to $60\,^\circ\text{C}$ and the pressure in the cabinet to 1 MPa under the condition of drying for more than 12 h Set aside.

### 2.2.2. Buckle Battery Assembly

Firstly, the dried negative electrode sheet, diaphragm, and pipette gun should be sent into the transition compartment of the glove box, the valve should be closed, and the vacuum should be pumped three times for battery assembly.

1. The negative shell will be in the glove box on the table.
2. Use clean tweezers; make sure that the clip of the lithium surface is smooth and concaves upward, and that the cathode shell is central.
3. The flat diaphragm should cover the lithium.
4. Using a pipetting gun, place 50 µL electrolytic liquid droplets in the surface of the diaphragm, so that they are fully infiltrated.
5. Then, place the cathode piece, coated with active substances, face down in the middle of the diaphragm.
6. Clip the gasket onto the cathode piece, and gently squeeze with forceps.
7. Clip the shrapnel and maintain the small mouth up. The positive shell should cover it.
8. Apply light pressure to ensure that the cover is on.
9. Using plastic tweezers, clip the batteries in the tableting machine, with pressure of 100 Pa; maintain the pressure for 3~5 s, and make sure that there is pressure to take out the battery assembly of good batteries, saving more than 8 h of waiting for the test at room temperature.

### 2.2.3. Battery Performance Test

1. Battery formation: At room temperature of 25 °C, the formation steps are as follows: Stand for 5 min; then, charge to 0.01 V at 0.1 A $g^{-1}$ current density and stand for 5 min; at 0.1 A $g^{-1}$ current constant, charge to 3.0 V, and then discharge at 0.1 A $g^{-1}$. The process should be repeated three times and then be ended (the current changes according to the quality of the active substance in the electrode).
2. The cell cycle: with 25 DHS C at room temperature and a charge and discharge voltage range of 0.01~3 V, the battery should be turned to the button cell, with cycles of 100~1000 times; when the charge and discharge current is set up according to turning it into the size of the battery charge given set, it can be divided into 0.1, with a 2 A $g^{-1}$ charge and discharge cycle.
3. Ratio test: this should come after the battery, respectively, in a 0.1, 0.2, 0.5, 1, and 2 A $g^{-1}$ charge and discharge test, under different ratios of each cycle for six laps.
4. The cyclic voltammetry test: This should be completed upon the assembly of the battery, at room temperature; set the voltage range to 0.01~3 V and the scan rate to 0.1 mV $s^{-1}$ in the cyclic voltammetry test, and analyze the material in the case of REDOX reaction voltage. Moreover, the properties of the materials can be judged according to the REDOX peaks' coincidence degree via the CV curves of multiple cycles.
5. AC impedance test: to assess EIS after the battery test, test the frequency in the range of 0.01–106 Hz, create the curve fitting using Zview software (Zview 3.2.0.49), and carry out electrochemical impedance analysis on the battery.

## 3. Results and Discussion

In order to further determine the structure of the prepared samples, firstly, several samples were analyzed using XRD (XPert Powder, Malvern Panalytical B.V., Almelo, The Netherlands). As shown in Figure 1a, all three samples showed characteristic peaks of $ZnFe_2O_4$ (PDF#22-1012). The diffraction peaks at 18.42°, 30.10°, 35.72°, 37.06°, 42.98°, 53.33°, 56.79°, 62.41°, and 73.77° correspond to (111), (220), and (317) of the $ZnFe_2O_4$ crystal. Additionally, there is a broad peak near 26.1°, but the peak intensity is weak. This is because the porous carbon formed by calcining SAP at 600 °C is the structural framework of the composite electrode. However, judging from the (002) main peak, the graphitization degree of the composite material is low, and the characteristic peak of the carbon matrix of the SAP-Fe/Zn&NC sample is not obvious, which is because it passes through $O_2$. However, the SAP-Fe/Zn&NC sample still maintains the complete phase of

the composite, while 30.5° and 35.0° are the (220) and (311) crystal planes of the $ZnFe_2O_4$ phase, respectively, which also confirms this point. Figure 1b shows the XPS (Thermo Fisher Scientific, Waltham, MA, USA) spectrum of SAP-Fe/Zn, SAP-Fe/Zn&N, and SAP-Fe/Zn&NC. C (284.8 eV), O (532.1 eV), Fe (725.1 eV, 726.1 eV), and Zn (1022.1 eV, 1046.1 eV) are given.

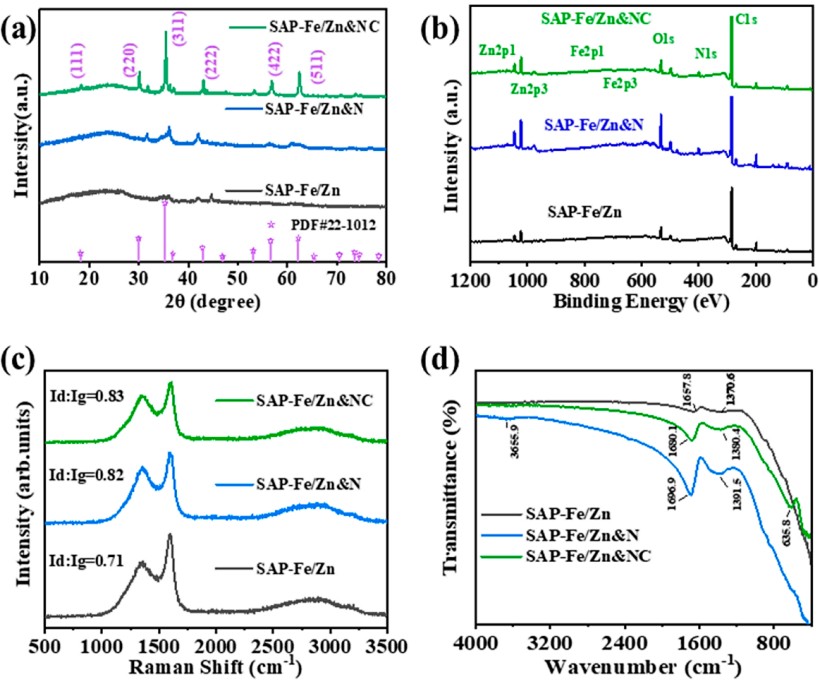

**Figure 1.** (**a**) XRD patterns of SAP-Fe/Zn, SAP-Fe/Zn&N, and SAP-Fe/Zn&NC samples; (**b**) XPS full spectrum; (**c**) Raman spectrum; (**d**) infrared spectrum.

In addition, it can be seen from the Raman spectrum (Figure 1c) that the two peaks of the three materials at 1343.9 and 1597.9 $cm^{-1}$ are related to the disordered (D) band and graphite (G) band of the carbon-based materials. Compared with the peak of SAP-Fe/Zn, the peak in SAP-Fe/Zn&NC shows an increased D/G intensity ratio (0.83), which is due to the decrease in the average size of the $sp^2$ domain, that is, the G band around 1585 $cm^{-1}$ vibrates in the plane corresponding to the $sp^2$ carbon atoms, and the D band around 1338 $cm^{-1}$ mainly reflects the structural defects in the graphite structure. The strength ratio ($I_D/I_G$) between SAP-Fe and Zn&N is 0.82, which indicates that the carbon disordered structure is dominant [22]. With SAP-Fe/Zn as a template, abundant porous structures are formed after doping N, and $Fe^{2+}$ ions are transformed into $Fe^{3+}$ in an oxygen atmosphere; then, the mechanical strength of the material is further improved via annealing, and then a stable three-dimensional porous carbon material is obtained via acid treatment. Figure 1d describes the FT-IR spectra of three composites. Among them, the bending vibration mode of the SAP-Fe/Zn sample is the hydrogen-bonded (H-O-H) hydroxyl group at 3655.9 $cm^{-1}$, while obvious fluctuation spectra appear at 1657.8 $cm^{-1}$, 1696.9 $cm^{-1}$, and 1680.1 $cm^{-1}$, representing SAP-Fe/Zn and SAP-Fe/Zn&N, respectively [23]. However, there is a band at 635.8 $cm^{-1}$, indicating that there is a metal oxygen bond in $ZnFe_2O_4$. Comparing the peaks of SAP-Fe/Zn and SAP-Fe/Zn&NC according to the corresponding characteristics, the interaction between the Fe/Zn nano-ions and N and C can be reflected, and the results show that there is a good interaction between $ZnFe_2O_4$ and SAP.

In order to determine the valence state and surface information of metal ions in the composite, X-ray photoelectron spectroscopy (XPS) was used to test them. The results are shown in Figure 2a, and the elemental composition of the Zn/Fe/SAP/N composite was further studied, corresponding to different forms of carbon atoms: oxygen-free carbon (C-C 284.8 eV) and carbon in the C-O group (epoxy or hydroxyl). As shown in Figure 2b,

but for different forms of O, after nitrogen doping and high-temperature oxidation, the peak of about 530.1 eV comes from the typical lattice O. The other peak is located at 532.3 eV, which is a defect site with low coordination of O, while the peak at 533.0 eV is chemisorbed O. At the same time, the formation of $ZnFe_2O_4$ is also affected by XPS measurement spectra O1s, Fe2p, and Zn2p XPS spectra. Figure 2c shows the Zn2p spectra of SAP-Fe/Zn&N and SAP-Fe/Zn&NC. The signals at 1044.58 and 1021.53 eV can be attributed to $Zn^{2+}Zn2p_{1/2}$ and $Zn2p_{3/2}$ [24]. In the high-resolution spectrum, it can be fitted as two peaks of Fe2p, as shown in Figure 2d. The peaks around 724.83 eV and 711.93 eV are related to the binding energies of $Fe2p_{1/2}$ and $Fe2p_{3/2}$, respectively. The observed photoelectron peaks of Zn2p and Fe2p are consistent with those of $Zn^{2+}$ and $Fe^{3+}$ in $ZnFe_2O_4$. However, due to the doping of nitrogen, the binding energy of SAP-Fe/Zn&N and SAP-Fe/Zn&NC changes, slightly shifting to a small angle [25].

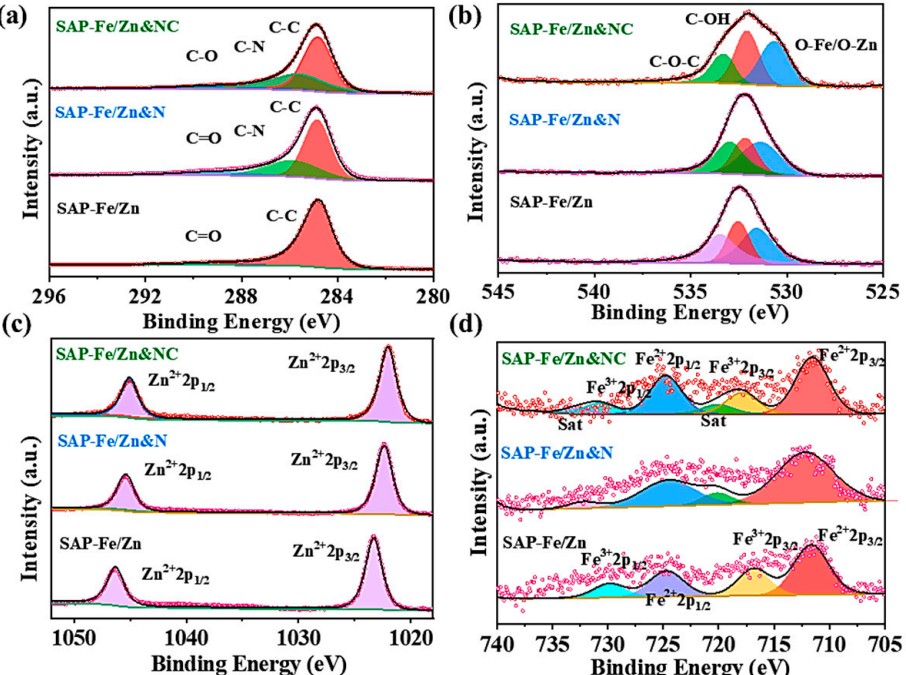

**Figure 2.** XPS fine spectrum of element (**a**) C of SAP-Fe/Zn, SAP-Fe/Zn&N, and SAP-Fe/Zn&NC samples; (**b**) XPS fine spectrum of O element; (**c**) XPS fine spectrum of Zn; (**d**) XPS fine spectrum of Fe element.

In order to explore the influence of N content on the surface of the materials after high-temperature oxidation, an X-ray photoelectron spectroscopy (XPS) test of three kinds of composites was carried out by fitting the fine figure, as shown in Figure 3a,b. The results show that the N1s peak is decomposed into three peaks at 398.6 eV, 401.5 eV, and 403.5 eV, which are attributed to pyridine, pyrrole, and graphite N atoms doped in the carbon material structure [26]. The negatively charged SAP carbon skeleton can adsorb positively charged $Fe^{3+}$ and $Zn^{2+}$ via electrostatic attraction in oxygen-containing groups, which can act as nucleation sites. On the other hand, the graphite nitrogen content of the SAP-Fe/Zn-NC samples decreased and the pyrrole nitrogen content increased after high-temperature oxidation. The pyrrole normally appears at 399.5 eV [27]. Furthermore, the binding energies obtained for the N-quaternary species, which normally appear at 401 eV, make me think that the graphitic nitrogen was overoxidized. The change in N content in the sample is shown in Table 1.

**Table 1.** Nitrogen content of SAP-Fe/Zn&N and SAP-Fe/Zn&NC samples.

| Types of N | | Graphitic-N | Pyridinic-N | Pyrrolic-N |
|---|---|---|---|---|
| Atomic (%) | SAP-Fe/Zn&N | 0.07 | 0.39 | 0.18 |
| | SAP-Fe/Zn&NC | 0.03 | 0.39 | 0.25 |

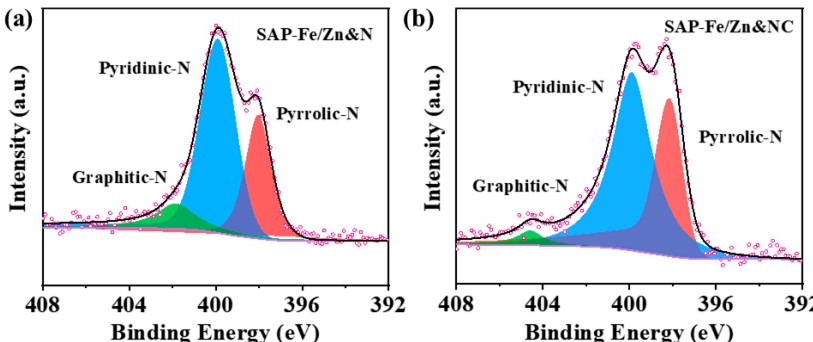

**Figure 3.** (**a**) XPS fine spectrum of N element in SAP-Fe/Zn&N sample; (**b**) XPS fine spectrum of N element in SAP-Fe/Zn&NC.

In order to further understand the morphology of the three composites, scanning electron microscope (SEM, Sigma 300, ZEISS, Oberkochen, Germany) analysis was carried out (Figure 4). It can be clearly seen from the image that the SAP carbon matrix in the SAP-Fe/Zn composite electrode (Figure 4a–c) is loaded with $ZnFe_2O_4$ nanoparticles with relatively uniform particle size. After being compounded with the SAP carbon material, the material presents nano-sized graded porous spheres with uniform particle size, which confirms the analysis of Raman spectrum and XRD. The assumption of uniform carbon coating is shown, but slight particle aggregation is observed, which can be explained by the interconnection of nanoparticles through carbon [28].

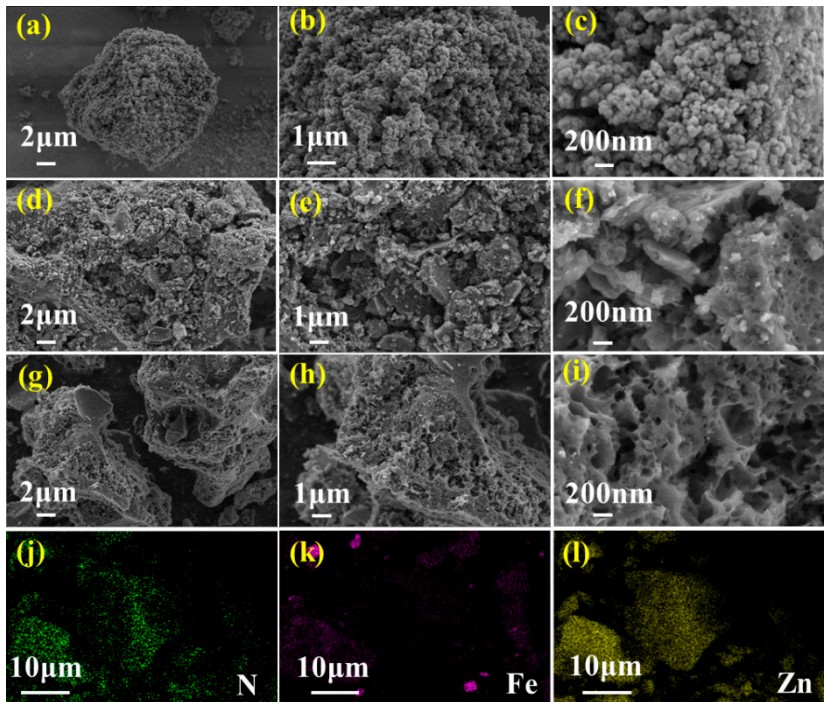

**Figure 4.** (**a**–**c**) SEM image of SAP-Fe/Zn sample; (**d**–**f**) SEM images of SAP-Fe/Zn&N samples; (**g**–**i**) SEM images of SAP-Fe/Zn&NC samples; EDS spectrum of (**j**–**l**) sap-Fe/Zn&NC sample.

After nitrogen doping, the samples of the SAP-Fe/Zn&N (Figure 4d–f) composites show that the nanoparticles are not highly aggregated. In addition, the aggregates of the $ZnFe_2O_4$ nanoparticles seem to be embedded in the carbon matrix, and most of the carbon matrix is staggered with irregular porous nanosheets, exposing many honeycomb pore defects. The distribution of $ZnFe_2O_4$ nanoparticles depends on the morphology and structure of the three-dimensional carbon skeleton. After that, it was oxidized in an $O_2$ atmosphere at 600 °C for one step, and then it was continuously annealed to obtain a three-dimensional porous composite with more regular morphology and more uniform pores, and the $ZnFe_2O_4$ particles were more densely connected with the carbon matrix (Figure 4g–i). The EDS spectrum also showed the uniform distribution of N, Fe, and Zn elements in the SAP-Fe/Zn&NC sample (Figure 4).

In order to better understand the morphology and structure of the composites, the prepared SAP-Fe/Zn, SAP-Fe/Zn&N, and SAP-Fe/Zn&NC were tested via TEM, as shown in Figure 5. In the TEM image of SAP-Fe/Zn, SAPs wrapped with some metal nanocrystals were observed (Figure 5a). By comparing this with the XRD image, the nanocrystals were identified as being $ZnFe_2O_4$, and the lattice spacing was captured as 0.250 nm using a high-resolution transmission electron microscope (Figure 5b), corresponding to the (311) crystal plane of the nanocrystal. Through the previous scanning image study, it was found that the microstructure of the SAP-Fe/Zn&N samples was irregular carbon clusters. Further transmission electron microscopy (TEM, Tecnai G2F 20, FEI, Morristown, NJ, USA) test results show that these carbon clusters are composed of randomly oriented irregular carbon layers (Figure 5c,d). For SAP-Fe/Zn&NC, as shown in Figure 5e,f, the microstructure also shows that the amorphous carbon structure is wrapped around the dark nanocrystals, and the nanocrystal structure is clear and complete. Combining SEM images with EDS-mapping (Smart EDX, ZEISS, Oberkochen, Germany), it can be observed that the distribution of Fe and Zn elements is similar, which is related to the encapsulated nanocrystals. The dark spots with uniform internal distribution and uniform size indicate the crystal state of $ZnFe_2O_4$. In addition, N has a similar distribution, which is because the carbon layer outside the nanocrystals is highly doped with N. From the above discussion, we can confirm that the combination of nitrogen-doped $ZnFe_2O_4$ and the SAP carbon matrix was completed in SAP-Fe/Zn, SAP-Fe/Zn&N, and SAP-Fe/Zn&NC composites, which verifies the previous test results and is beneficial for improving the lithium storage performance of electrode materials.

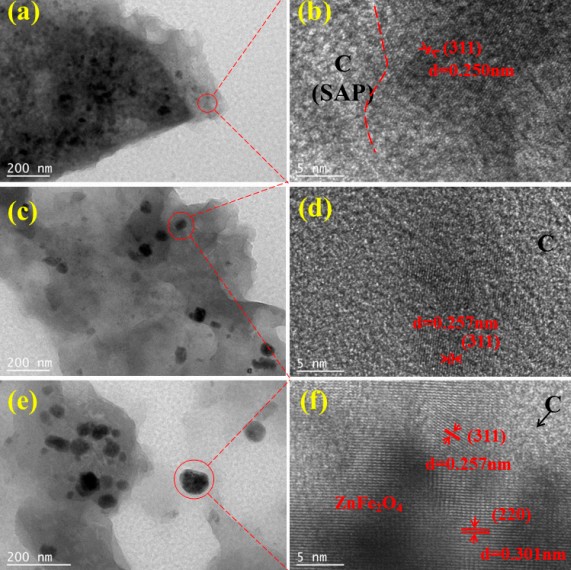

**Figure 5.** (**a**,**b**) TEM spectrum of sap-Fe/Zn sample; (**c**,**d**) TEM images of SAP-Fe/Zn&N samples; TEM spectra of (**e**,**f**) sap-Fe/Zn&NC samples.

Figure 6a shows the CV diagram of the SAP-Fe/Zn electrode scanned at 0.1 mV s$^{-1}$. During the first cathodic scanning, a large reduction peak of about 0.6 V was observed, corresponding to the irreversible transformation from $ZnFe_2O_4$ to $LiZn/Fe/Li_2O$ after lithium absorption. The overlapping peak of this peak was related to the formation of solid electrolyte interface (SEI) layer, and the potential of this phenomenon shifted to a small potential in SAP-Fe/Zn&N and SAP-Fe/Zn&N because of the acceleration of the N element. As shown in Figure 6b,c, in addition, during the second to fourth cathode scanning, the position of the reduction peak shifted to a higher potential, indicating that the reduction of $Fe_2O_3$ was carried out in two steps, while the corresponding anode peaks were at 1.56 V and 1.57 V. During the second to fourth scanning, the oxidation peak and reduction peak almost overlapped, indicating that the reversible conversion reaction between the $ZnO/Fe_2O_3$ and $LiZn/Fe/Li_2O$ lattice was good [29]. From the comparison of the first cyclic voltammetry curves of the three electrodes (Figure 6d), it can be observed that SAP-Fe/Zn&N and SAP-Fe/Zn&NC electrodes have larger integrated areas than SAP-Fe/Zn electrodes, which shows that they can provide more reversible capacity [30].

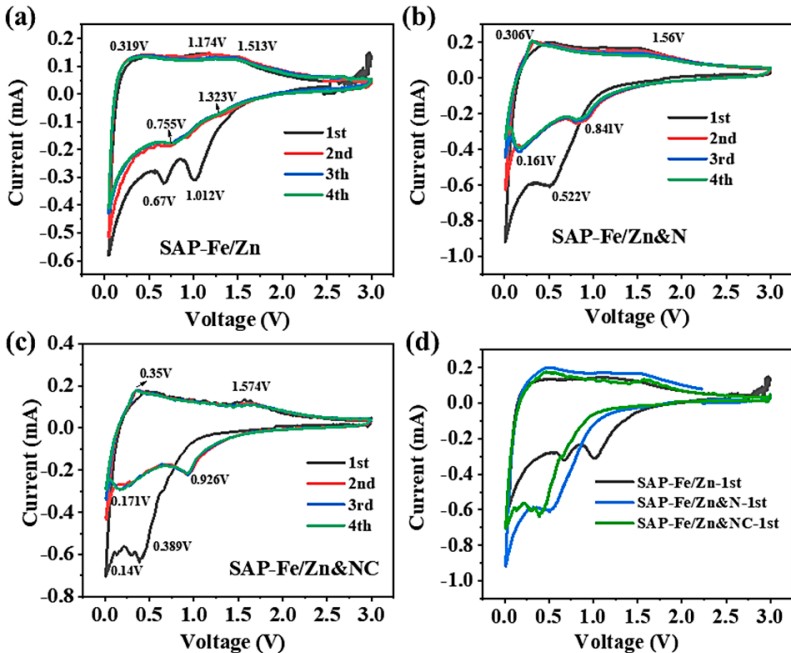

**Figure 6.** (**a**) Cyclic voltammetric curve of SAP-Fe/Zn electrode; (**b**) cyclic voltammetric curve of SAP-Fe/Zn&N electrode; (**c**) cyclic voltammetric curve of SAP-Fe/Zn&NC electrode; (**d**) comparison of the first-cycle voltammetric curves of SAP-Fe/Zn, SAP-Fe/Zn&N, and SAP-Fe/Zn&NC electrodes.

On the contrary, in the process of lithium removal, only the voltage plateau was observed at 1.5~3 V. During anodic scanning, the sum of the two peaks was between 1.55 V and 1.574 V, which is related to the oxidation of Zn and Fe to $Zn^{2+}$ and $Fe^{2+}$. Under the optimum conditions, $Fe^{2+}$ can be further reversibly oxidized to $Fe^{3+}$, as shown in Equations (1)–(6) [31].

$$ZnFe_2O_4 + xLi^+ + xe^- \rightarrow Li_xZnFe_2O_4 \tag{1}$$

$$Li_xZnFe_2O_4 + (8-x)Li^+ + (8-x)e^- \rightarrow ZnO + 2FeO + 4Li_2O \tag{2}$$

$$ZnO + Li^+ + e^- \leftrightarrow LiZn \tag{3}$$

$$ZnO + Li_2O \leftrightarrow ZnO + 2Li^+ + 2e^- \tag{4}$$

$$2FeO + 2Li_2O \leftrightarrow 2FeO + 4Li^+ + 4e^- \tag{5}$$

$$2FeO + Li_2O \leftrightarrow Fe_2O_3 + 2Li^+ + e^- \tag{6}$$

Figure 7 shows the charge (lithium removal)–discharge (lithiation) curves of the three electrodes at 100 mA g$^{-1}$ in the first three cycles. The capacity of several electrode materials is calculated according to the total mass of ZnFe$_2$O$_4$ and SAP carbon materials. Combined with the data of constant current charge and discharge in Table 2, the first charge and discharge capacities of SAP-Fe/Zn are 728.0 and 1165.5 mAh g$^{-1}$, respectively. After nitriding, the first charge and discharge capacities of the SAP-Fe/Zn&N electrode are improved, reaching 840.2 and 1056.9 mAh g$^{-1}$, respectively, and the first coulombic efficiency is as high as 79.5%. The first charge and discharge capacities of SAP-Fe/Zn&NC are 1101.6 mAh g$^{-1}$ and 1712.5 mAh g$^{-1}$. According to the reaction (ZnFe$_2$O$_4$ → 4Li$_{0.5}$ZnFe$_2$O$_4$ → 4Li$_2$ZnFe$_2$O$_4$ → Li$_2$O + Li–Zn + Fe), the theoretical capacity of ZnFe$_2$O$_4$ is estimated to be 1000 mAh g$^{-1}$. The extra capacity provided by the SAP-Fe/Zn&N and SAP-Fe/Zn&NC electrodes may be attributed to the synergistic effect between ZnFe$_2$O$_4$ nanocrystals and the carbon anode [32].

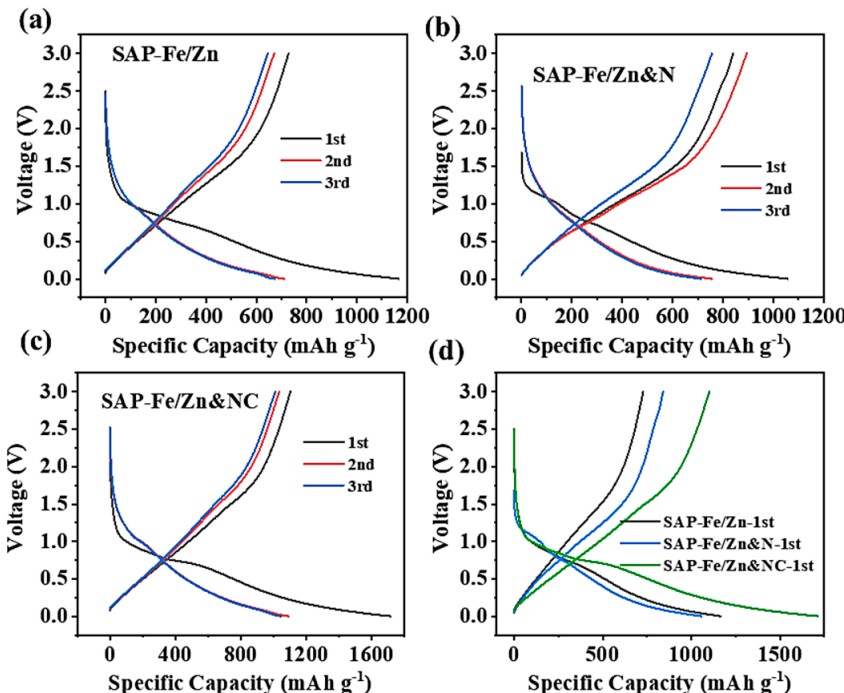

**Figure 7.** (**a**) Charge–discharge curve of SAP-Fe/Zn electrode; (**b**) charge–discharge curve of SAP-Fe/Zn&N electrode; (**c**) charge–discharge curve of SAP-Fe/Zn-NC electrode; (**d**) comparison of the first charge–discharge curves of the SAP-Fe/Zn, SAP-Fe/Zn&N, and SAP-Fe/Zn&NC electrodes.

**Table 2.** Constant current charge and discharge data of SAP-Fe/Zn, SAP-Fe/Zn&N, and SAP-Fe/Zn&NC electrodes.

| Sample | First Discharge-Specific Capacity (mAh g$^{-1}$) | First Charge-Specific Capacity (mAh g$^{-1}$) | First Charge and Discharge Efficiency (%) |
|---|---|---|---|
| SAP-Fe/Zn | 1165.5 | 728.0 | 62.5 |
| SAP-Fe/Zn&N | 1056.9 | 840.2 | 79.5 |
| SAP-Fe/Zn&NC | 1712.5 | 1101.6 | 64.3 |

On the one hand, due to nitrogen doping, many $ZnFe_2O_4$ nanocrystals are uniformly dispersed in the composite electrode, which leads to good electrolyte wetting and a high utilization rate of active particles. On the other hand, the existence of $ZnFe_2O_4$ nanocrystals has a certain catalytic effect, which makes the carbon matrix easier to be exposed to electrolytes and provides many active sites for lithium-ion storage. Additionally, in low-dimensional materials, the increase in the capacity of the composite electrode is also related to the increase in the quantum capacitance. The irreversible capacity generated by the three electrodes can be attributed to the formation of the SEI layer, as described above.

Figure 8a shows the Nyquist diagram of the activated electrodes of SAP-Fe/Zn, SAP-Fe/Zn&N, and SAP-Fe/Zn&NC. The Nyquist diagram consists of two partially overlapping semicircles in the high-frequency and intermediate-frequency regions and diagonal lines in the low-frequency region. The first semicircle is related to the resistance of lithium-ion transmission through the SEI layer, the second semicircle is related to the resistance of charge transfer, the diagonal line is related to the diffusion of lithium-ion in bulk materials, the intercept with respect to the X-axis in the high-frequency region is related to the resistance of the system, and, ultimately, the electrolyte includes the conductivity of the material itself, which is drawn by the equivalent circuit shown in the inset in Figure 8b. In the equivalent circuit, represents electrolyte resistance, $R_f$ and $Q_1$ represent SEI layer resistance and dielectric relaxation capacitance, $R_{ct}$ and $Q_2$ represent charge transfer resistance and double-layer capacitance, and $W_0$ is diffusion resistance [33]. Table 3 shows the impedance fitting results of three kinds of electrodes.

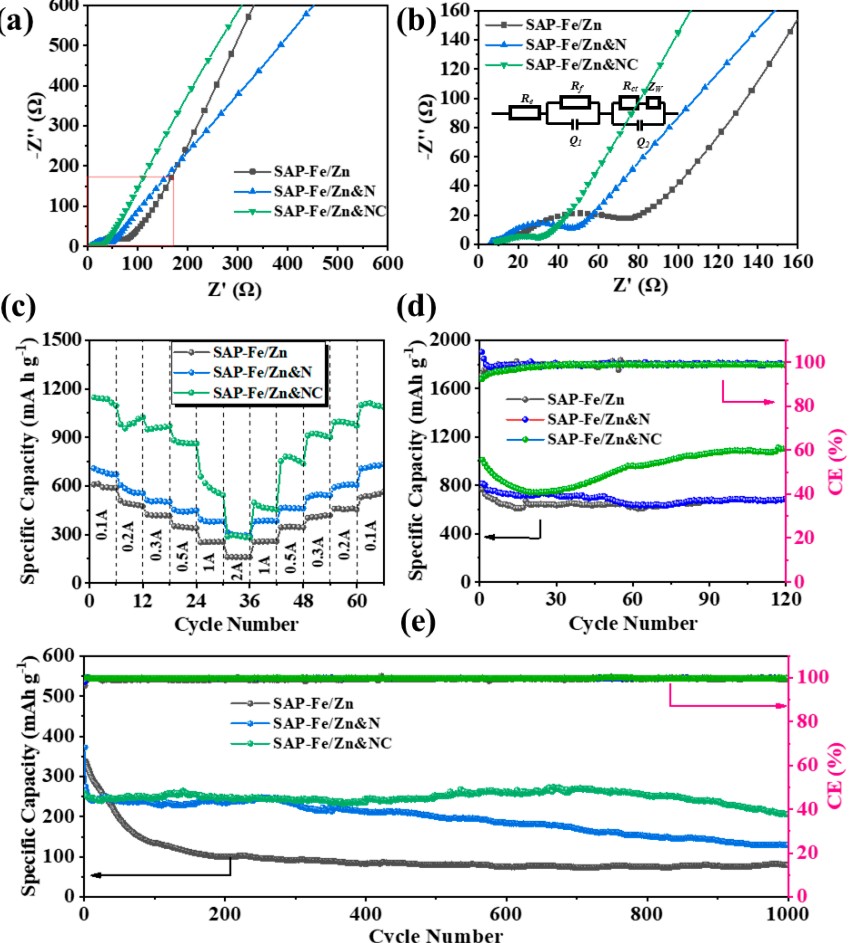

**Figure 8.** (**a**) EIS curves of SAP-Fe/Zn, SAP-Fe/Zn&N, and SAP-Fe/Zn&NC electrodes; (**b**) EIS curve (equivalent circuit); (**c**) rate performance curve; (**d**) 0.1 A g$^{-1}$ cycle curve comparison; (**e**) comparison of 2 A g$^{-1}$ cycle curves.

**Table 3.** Impedance fitting data of SAP-Fe/Zn, SAP-Fe/Zn&N, and SAP-Fe/Zn&NC electrodes.

| Sample | $R_e$ (ohm) | $R_{ct}$ (ohm) | $W_0$ (ohm) |
|---|---|---|---|
| SAP-Fe/Zn | 10.69 | 89.0 | 0.37 |
| SAP-Fe/Zn&N | 6.02 | 47.34 | 0.62 |
| SAP-Fe/Zn&NC | 6.20 | 35.13 | 0.70 |

The charge cycle curves and coulomb efficiency curves of the SAP-Fe/Zn composites and nitrogen-doped SAP-Fe/Zn&N and SAP-Fe/Zn&NC composites are shown in Figure 8d,e, respectively. Among them, the SAP-Fe/Zn&NC composite showed excellent cycle stability, the capacity of the SAP-Fe/Zn&NC composite electrode was still 1102.1 mAh g$^{-1}$ after 120 cycles at a current density of 0.1 A g$^{-1}$, and the capacity retention rate was as high as 109.1%. At a high current density of 2 A g$^{-1}$, the composite electrode of SAP-Fe/Zn&NC had high reversible capacity after 1000 cycles. It finally stabilized at 205.6 mAh g$^{-1}$, and the capacity retention rate was 80.7%, which was much higher than that of SAP-Fe/Zn or SAP-Fe/Zn&N with 23.5% and 47.4%. The specific cycle data are shown in Tables 4 and 5. Other transition metal oxides also show similar behavior (gain extra capacity), which is related to the (partial) reversible formation of a polymer layer on the particle surface [34]. The excellent cycle stability of $ZnFe_2O_4$ benefits from its structure, which provides an interesting combination of alloying and carbon layer transformation mechanisms. Compared with SAP-Fe/Zn, the prepared $ZnFe_2O_4$/C composite electrode also showed a good rate of performance. As shown in the study of the charge/discharge rate performance in Figure 8c, when the charge/discharge rate is gradually increased from 0.1 A g$^{-1}$ to 2 A g$^{-1}$, the reversible capacity is reduced from 1147 mAh g$^{-1}$ to 288.5 mAh g$^{-1}$. When the rate finally returns to 0.2 A g$^{-1}$, the reversible capacity of 1099.3 mAh g$^{-1}$ can be recovered. In contrast, the N-doped SAP-Fe/Zn electrode shows a low capacity and a sharp decline in capacity at a high specific current. Tables 4 and 5 show the specific data.

**Table 4.** Comparison results of cycle performance of three electrodes at current density of 0.1 A g$^{-1}$.

| Sample | Specific Capacity for First Charge (mAh g$^{-1}$) | After 120 Cycles of Charge-Specific Capacity (mAh g$^{-1}$) | Cycle Retention Rate (%) |
|---|---|---|---|
| SAP-Fe/Zn | 765.8 | 679.3 | 88.7 |
| SAP-Fe/Zn&N | 814.7 | 682.8 | 83.8 |
| SAP-Fe/Zn&NC | 1010.6 | 1102.1 | 109.1 |

**Table 5.** Comparison results of cycle performance of three kinds of electrodes at a current density of 2 A g$^{-1}$.

| Sample | Specific Capacity for First Charge (mAh g$^{-1}$) | After 1000 Cycles of Charge-Specific Capacity (mAh g$^{-1}$) | Cycle Retention Rate (%) |
|---|---|---|---|
| SAP-Fe/Zn | 337.4 | 79.4 | 23.5 |
| SAP-Fe/Zn&N | 274.4 | 130.2 | 47.4 |
| SAP-Fe/Zn&NC | 254.7 | 205.6 | 80.7 |

## 4. Conclusions

To sum up, carbon composites coated with $ZnFe_2O_4$ (SAP-Fe/Zn) were synthesized via sol–gel and the two-step pyrolysis strategy, nitrogen-doped $ZnFe_2O_4$ carbon composites (SAP-Fe/Zn&N and SAP-Fe/Zn&NC) were synthesized via the simple template removal method, and their energy storage functions as negative electrodes of lithium-ion batteries were discussed.

For SAP-Fe/Zn composites, $ZnFe_2O_4$ nanoparticles show uniform loading via the high-temperature carbonization of polymers and crystallization of inorganic elements. Under the current density of 0.1 A g$^{-1}$, the reversible capacity reached 679.3 mAh g$^{-1}$

after 120 cycles, and showed an excellent rate of performance. After nitriding, the capacity of the SAP-Fe/Zn&N electrode was improved, the first charge–discharge capacity was 840.2 and 1056.9 mAh g$^{-1}$, respectively, and the first coulombic efficiency was as high as 79.5%. SAP-Fe/Zn&NC, as the anode material of the lithium-ion battery, has a first discharge/charge capacity of 1712.5/1106.1 mAh g$^{-1}$, a coulomb efficiency of 64.3%, a reversible capacity of 205.6 mAh g$^{-1}$ after 1000 cycles at a high current density of 2 A g$^{-1}$, a cycle retention rate of 80.7%, and a good rate of performance. The modification of electrochemical performance is attributed to its continuous structural skeleton and uniform surface pores, which is beneficial to the reversible lithium/lithium removal process, and nitrogen has strong electronegativity, which makes it possible to greatly improve the activity of nanoparticles in a carbon matrix. This clear structure and morphology promoted the transport of Li$^+$, enhanced the effective contact area with the electrolyte, and provided abundant active sites. The enhanced electrochemical performance can be attributed to the one-dimensional nanostructure and the shortened diffusion path, which ensured the full conversion reaction in the lithium–lithium process between Zn, Fe, and Li$^+$, relieved the volume expansion, and prevented the crushing/aggregation when circulating for a long time at a high current density. Therefore, we think that SAP-Fe/Zn&NC has great potential as an anode material for lithium-ion batteries. Transition metal oxides with a carbon material of the composite aim to improve the material, and they are one of the main means of electrochemical properties. The synergistic effect between them can effectively improve the performance of the composite material, so that it has excellent electrochemical performance as the anode of the lithium/sodium ion battery. However, the preparation of composite materials is complicated, which is not conducive to large-scale production. On the other hand, the reaction between the $ZnFe_2O_4$/N-doped composite and intercalation during lithium storage needs further investigation.

**Author Contributions:** Data curation, L.W., N.X. and Z.Y.; Writing, L.W.; Formal analysis, B.L. and C.Y.; Investigation, H.B. and H.D.; Methodology, J.X.; Conceptualization, X.R. and B.D. All authors have read and agreed to the published version of the manuscript.

**Funding:** The research was financially supported by National Natural Science Foundation of China (21865012, 52202064), Education Department Project Fund of Jiangxi Province (GJJ190427), Ganzhou innovative talent project, the Program for Excellent Young Talents, JXUST, Science and Technology Department of Henan Province (222102230054), Luoyang major science and technology project, New Energy Electric Vehicles High-Voltage Components Inspection and Testing Public Service Platform, Henan Province Education Department of Key Scientific Research Project in Colleges and Universities (21B430012, 23B430012).

**Institutional Review Board Statement:** Not applicable.

**Informed Consent Statement:** Not applicable.

**Data Availability Statement:** Not applicable.

**Acknowledgments:** Not applicable.

**Conflicts of Interest:** The authors declare no conflict of interest.

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
