# Peer review of "Preparation of Nitrogen-Doped ZnFe2O4-Modified Carbon Composite and Its Collaborative Energy Storage Mechanism"

_coatings, doi:10.3390/coatings13061126_

Round 1

Reviewer 1 Report

In this research Wang et al. synthesized carbon-based composites modified with ZnFe2O4 and nitrogen. Electrochemical tests showed that addition of nitrogen and calcined nitrogen significantly improves specific capacity of materials. To my mind, the paper is deserves publication but some moment should be revised.

1) Introduction suffers from the lack of review of modern papers devoted to carbon composites modified with ZnFe2O4. For example, following paper can be cited and reviewed:

https://doi.org/10.1021/acsami.8b20305

https://doi.org/10.1021/acsnano.5b07806

https://doi.org/10.1039/C5TA00805K

https://doi.org/10.1039/C7TA02726E

https://doi.org/10.1038/srep43116

https://doi.org/10.1039/C8NR08611G

In the Conclusion section authors should state what new result in this field they achieved (on the base of comparison with these papers)

2) Can be the enhacement of capacity connected to growth of quantum capacitance part as it always happens for low-dimensional materials. Little discussion of this moment can be useful

Reviewer 2 Report

Manuscript Number: coatings-2387618

Title: Preparation of nitrogen doped ZnFe2O4 modified carbon composite and its collaborative energy storage mechanism

This paper reported on the synthesize, characterize and performance evaluation of carbon composites coated with ZnFe2O4 as negative electrode of lithium-ion battery. In general, the topic is of interest to the energy storage community, especially for lithium-ion battery technology. Overall, the finding data, discussion, and the goal of the present paper are good, and its can be considered for the Coatings. However, several part need to be improved for this current manuscript and authors should be paying special attention to the following comments;

1. Abstract should be written in concise and precisely. The quantitative and qualitative finding should be highlighted.

2. Introduction: It is not so clear the novelty of the study. It is on the synthesize method, i.e., two-step pyrolysis strategy, or the type of used material. Please clarify in the introduction section.

3. Experimental part: Samples characterization and performance evaluation methods should be described in this part.

4. Conclusion: Abstract should be written in concise and precisely.

Reviewer 3 Report

The manuscript entitled “Preparation of nitrogen-doped ZnFe2O4 modified carbon composite and its collaborative energy storage mechanism” has been submitted by the authors. Some issues to be addressed will improve the quality of the manuscript. Therefore, I recommend this work could be published after the major revision

1.      The manuscript presents an interesting approach for developing a nitrogen-doped ZnFe2O4 modified carbon composite. The results show promising performance as a potential candidate for various applications, such as energy storage and conversion. However, some parts of the manuscript need further clarification and improvement to strengthen the arguments.

2.      The author should write down the novelty and result in abstract.

3.      The introduction section provides a good background on the topic, but it lacks a clear research question and hypothesis. The authors should clearly state their research objectives and the expected outcomes of the study.

4.      The methods section is comprehensive and well-written. However, there are some ambiguities in the experimental setup that need clarification. For instance, the authors should specify the particle size and morphology of the ZnFe2O4 nanoparticles and provide more details on the synthesis of the carbon precursor.

5.      The results section presents a detailed analysis of the physical and chemical properties of the composite. However, the authors should include more discussion and interpretation of the results, particularly in relation to the research question and hypothesis. Also, the authors should provide more details on the stability and reproducibility of the composite.

6.      The discussion section provides a good summary of the findings and their potential implications. However, the authors should avoid general statements and provide more specific conclusions based on the results. Also, the authors should highlight the limitations of the study and suggest possible directions for future research.

7.      The manuscript is well-written and organized. However, there are some grammatical and typographical errors that need correction. The authors should proofread the manuscript carefully before submission.

8.      The references section provides a good selection of relevant literature. However, some of the references are outdated or incomplete. The authors should update the references and ensure that they are cited correctly in the manuscript.

Reviewer 4 Report

In the current work, the authors have synthesized a nitrogen-doped ZnFe2O4 carbon composite material employing super polymer adsorbent resin (SAP) as a template for carbon material synthesis. The material was subsequently employed as a negative electrode in a lithium-ion-based battery. The work in itself shows low novelty in terms of the material evaluated, but the use of an N-doping process of the carbon material could be interesting. However, there are important comments that should be solved to ensure the acceptance of the work:

1. The authors have to check minor mistakes and also some "informal" language employed in the manuscript, but the nomenclature employed for the materials synthesized is not clear.

2. In the XPS spectra, there are a lot of concerns about the results obtained. Firstly, Zn is a complicated element, especially, when oxidized species of Zn2+ want to be identified. I see shifting to lower energy bindings for Zn signal, which is quite notorious in the XPS spectra when the N species are incorporated and the carbonatization takes place, I recommend to the authors check the Auger of Zn, because that one is more sensitive to the change in the oxidation state. Actually, the heat treatment can produce a reduction of Zn by the CO produced. Furthermore, it is really hard to agree that the solid ZnFe2O4 is generated taking into account that the relative concentration of Zn and Fe in the XPS is quite different, despite being more iron in the stoichiometry of the compound. Actually, the compositional mapping in SEM shows clearly segregation of Zn in areas that do not match with the Fe, suggesting that there is the presence of ZnO, Zn0, or iron oxides (Fe3O4 and Fe2O3). Is it possible to demonstrate iron oxides and ZnO are not present? The second thing is related to the N species, I disagree with the values of energy binding, for the pyrrole which normally appears at 399.5 (References: https://doi.org/10.1002/ange.202000936, https://doi.org/10.1016/j.carbon.2022.05.032). Furthermore, the binding energies obtained for the N-quaternary species, which normally appears at 401 eV, make me thing that the graphitic nitrogen is overoxidized and should be specified in the manuscript.

3. There is a complete lack of explanation of the electrochemical characterization of no electrolyte, conditions of the cell, potential reference, electrolyte, and conditions of impedance. It is MANDATORY to include this information and the equipment employed. Please, reagents and suppliers are also useful to understand. It is impossible to make an accurate evaluation of this section based on the CV, Charge-discharge profiles, and EIS.

4. Can the authors explain why there is not a plateau region in the discharging step for any samples? There are oxidative transition steps that are involved in the process that has not been considered, such as in the case of iron.

5. It is called the Nyquist diagram, not Inquest. Line 275.

6. In Table 3, the authors show the values of the resistance elements of the equivalent circuit chosen, however, I would like to clarify something for the statement from lines 279-281, the intercept with the X-axis at high-frequency region is not the resistance of the electrolyte solution, it is related with the electrical resistance of the system, eventually the electrolyte includes, but also the conductivity of the material in itself. So, the statement is false, another way to prove it is, if you are using the same electrolyte why does the resistance of this one change? It is obvious that the carbonization produces some graphitization process which improves the resistance.

7. Please, the literature is abundant in this material, can the authors include and compare the results with some reported in the literature? Example: https://doi.org/10.1021/acs.chemmater.7b00467, https://doi.org/10.3390/app112411713

8. Is there any measurement of the specific surface area of the materials obtained as well as the ex-situ conductivity measurements?

9. The cyclic voltammetry response in Figure 3 shows two main features that are concerning me, mainly related to the stripping wave of Li+, which is not well observed as in other lithium-ion systems as well as the same material. The Li+ reduced is effectively released afterward in the oxidation wave? In the same order of ideas, can the reduction peak at 1 V approx. be related to a reduction process of the iron or zinc? Apparently, overlap or disappear with doping and carbonization but why? Is there any explanation? Is the charge of the reduction and oxidation the same?

Round 2

Reviewer 3 Report

The author solve all comment carefully, I recommended to accept in present form. 

Author Response

Thank you very much.

Reviewer 4 Report

I really appreciate the response from the Authors and changes, but there is no response in terms of the Auger for Zn species or an explanation of the possibility of segregation of Zn and Fe in other oxide forms. It is a clear valid question that has not been answered by the Authors and a suggestion to improve the quality of the work.
